# Unexpected High Species Diversity of *Mesolycus* Gorham (Coleoptera, Lycidae) from China, with a Preliminary Investigation on Its Phylogenetic Position Based on Multiple Genes [note 1]

**DOI:** 10.3390/insects13121171

**Published:** 2022-12-17

**Authors:** Haoyu Liu, Ruolan Du, Ya Kang, Xueying Ge, Xingke Yang, Yuxia Yang

**Affiliations:** 1Key Laboratory of Zoological Systematics and Application, School of Life Science, Institute of Life Science and Green Development, Hebei University, Baoding 071002, China; 2Key Laboratory of Zoological Systematics and Evolution, Institute of Zoology, Chinese Academy of Sciences, Beijing 100101, China

**Keywords:** net-winged beetles, Dilophotini, Macrolycini, phylogeny, taxonomy

## Abstract

**Simple Summary:**

The East Palaearctic fauna of net-winged beetles is very rich, but poorly studied in China. As an important faunistic component of the East Palaearctic fauna, the detailed taxonomic studies of Lycidae from China are necessary for understanding the evolution of this important biodiversity hotspot. More importantly, it is a region where some lycid lineages occur only there and in an adjacent part of the Oriental Region, such as *Mesolycus* Gorham, 1883. The genus remains controversial in its systematic placements but has never been rigorously tested due to unavailability of sampling material. In the present study, the genus *Mesolycus* was reviewed, with an emphasis on the species diversity from the Chinese fauna. Meanwhile, multiple genetic data of this genus were obtained and included in the phylogeny of lycid beetles for the first time. As a result, it is quite surprising that the number of the Chinese *Mesolycus* species is increased from 6 to 13, which makes China stand out in the species diversity on a world scale. Meanwhile, the statuses of some previously known species were clarified, and the systematic placement of *Mesolycus* was verified. This study makes us better understand the species diversity and phylogenetic position of *Mesolycus*.

**Abstract:**

The lycid genus *Mesolycus* Gorham, 1883 is mainly distributed in East Palaearctic and Indochinese regions, but poorly studied in China; moreover, its phylogenetic placement remains controversial but has never been rigorously tested. In this study, *Mesolycus* was reviewed and its placement within Lycidae was tested based on a multilocus phylogeny (*cox1*, *nad5*, *cox2* and *Lrna*) by both ML and BI analyses. The reconstructed phylogenies show that *Mesolycus* is a consistently recovered sister to *Dilophotes* Waterhouse, 1879, and they form a monophyletic clade which is well supported. This suggests that *Mesolycus* definitely belongs to Dilophotini rather than to Macrolycini of Lycinae. Besides, three species originally described or placed in *Dilophotes* are transferred to *Mesolycus*, including *M. atricollis* (Pic, 1926) comb. n., *M. particularis* (Pic, 1928) comb. n. and *M. pacholatkoi* (Bic, 2002) comb. n. Four new species are discovered in China, including *M. shaanxiensis* sp. n., *M. dentatus* sp. n., *M. breviplatus* sp. n. and *M. varus* sp. n. Two species, *M. murzini* Kazantsev, 2004 and *M. rubromarginatus* Kazantsev, 2013, are recorded from China for the first time. A key for the identification of all *Mesolycus* species is provided. China was revealed as the region with the highest species diversity of this genus.

## 1. Introduction

Lycidae, or net–winged beetles, is a moderately large beetle family, with about 4600 species hitherto known in the world [1,2]. Many species occur in small ranges and their diversity has been commonly underestimated [3,4,5].

The East Palaearctic fauna of net–winged beetles is very rich [6], but poorly studied in China. It is a region where some lineages occur only there and in an adjacent part of the Oriental Region, such as Macrolycini and Lyponiini [1], which have the highest diversity in East Palaearctic, and very probably expanded their ranges to the Oriental region [7]. Meanwhile, some lineages, such as Dilophotini, which has more species in the Oriental region than those of the East Palearctic, may be present because they are in a converse dispersal route [2,8,9,10]. Therefore, as an important faunistic component of the East Palaearctc fauna, the detailed taxonomic studies of the Lycidae from China are necessary for understanding the evolution of this important biodiversity hotspot [11].

The lycid genus *Mesolycus* Gorham, 1883 was proposed for the Japanese species *Eros atrorufus* Kiesenwetter, 1879. Although Nakane [6] has already shown the close relationship between *Mesolycus* and *Dilophotes* Waterhouse, 1879, he did not formally synonymize them, which was done later by Bocak and Bocakova [12]. Until Kazantsev [9], who examined the type species for each genus, revalidated *Mesolycus* from synonym with *Dilophotes*, he meanwhile synonymized *Flabellodilophotes* Pic, 1921 and *Dilophotes* (*Biphilodes*) Kazantsev, 2000 with *Mesolycus*. Up to date, a total of 30 species of *Mesolycus* were recorded from the East Palaearctic and Oriental regions, of which six species occurred in China [8,9,10,13]. However, since it was redefined by Kazantsev [9], no systematic work has been done on this genus, and some species remain unclear in their placements; moreover, multiple species remain unknown particularly in China.

The genus *Mesolycus* was redefined and the diagnostic characteristics were summarized well by Kazantsev [9], which is recognizable by a blunt angle between the vertex and frons planes, serrated antennae hardly reaching over the elytral middle in both sexes, absence of wedge cells in metathoracic wings, plantar pads usually being present on tarsomeres 1 and 2 (absent in *M. shelfordi* (Bourgeois, 1906)), and with symmetrical aedeagus including the inner sac structures.

Due to the poorly resolved taxonomy of the group the systematic placement of *Mesolycus* is uncertain. This genus, once a junior synonym of *Dilophotes*, was placed in the tribe Macrolycini Kleine, 1929 [8,10,12,14]. The name Dilophotini was erected for *Dilophotes* by Kleine [15] and synonymized to Macrolycini by Bocak and Bocakova [16], but the former was revalidated as a subtribe of the latter by Kazantsev based on a morphologically phylogenetic analysis [9]. In the latter study, *Mesolycus* together with *Macrolycus* Waterhouse, 1878 and *Calopteron* Kazantsev, 2004 were placed in Macrolycina, and *Dilophotes* was solely included in Dilophotina, which was soon resurrected into a separate tribe based on another hypothesis of morphological phylogeny [17]. After that, *Mesolycus* was noted to be a member of Dilophotini rather than Macrolycini by Bocak & Bocakova [1], but no material was included in their multigene phylogenetic analysis. However, this placement was not agreed by Kazantsev [13], who insisted that *Mesolycus* was a member of Macrolycini. Although another molecular phylogeny of Lycidae [18] was reconstructed recently, no *Mesolycus* sample was included in the analysis. Therefore, the systematic status of *Mesolycus* remains controversial at present.

In our recent study, we obtained some material of *Mesolycus* from China. After our examination, we discovered some new species and describe them herein. Given the narrow geographic distribution of most lycid beetles, it seems plausible that many species are vanishing without ever having been collected due to habitat loss. In light of the biodiversity crisis and accelerating rates of biological extinction, taxonomic work and targeted collecting (with specimens suitable for both morphological and molecular work) remain essential for the future of systematic studies. Luckily, we successfully sequenced the mitochondrial genomes of two species, which allowed us to investigate the phylogenetic position of *Mesolycus*. The present study is another contribution to the knowledge of *Mesolycus*, with an emphasis on exploring the species diversity of this lycid group in the Chinese fauna.

## 2. Materials and Methods

### 2.1. Materials

The studied material is preserved in the Institute of Zoology, Chinese Academy of Sciences, Beijing, China (IZAS) and the Museum of Hebei University, Baoding, China (MHBU). The sampling material of two *Mesolycus* species was preserved in 100% ethanol at −20 °C and catalogued in the voucher collection (Nos. CAN-25 and 2CA179) at MHBU. As these species were considered non–protected invertebrates, no permits were required for this study.

### 2.2. Morphological and Geographical Studies

The specimens were softened in water, and the male genitalia were dissected, then cleared in 10% NaOH solution, examined and photographed in glycerol, and finally glued on a paper card for permanent preservation. Habitus photos were taken by a Canon 7D camera and aedeagi by Leica M205A stereomicroscope; multiple layers were stacked using Combine ZM (Helicon Focus 5.3) and edited in Adobe Photoshop CS3.10.0.1. The aedeagi was illustrated and colored in Inkscape 1.0.2-2 (e86c870879, 2021-01-15).

The measurements were taken with Image J 1.50i (NIH, USA). Body length was measured from the anterior margin of the clypeus to the elytral apex, and the width across the humeral part of elytra. Pronotal length was measured from the middle of the anterior margin to the middle of the posterior margin and the width across the widest part of the pronotum. Eye diameter was measured at the widest point and the interocular distance was taken at the point of the minimum.

The distribution information was collected from the original publications and the material studied in this study (Appendix A). The distribution map was prepared by the ArcMap 10.8 and edited in Adobe Photoshop CS3.10.0.1.

### 2.3. DNA Extraction, Sequencing and Annotation

Total DNA was extracted from the thoracic muscle using a DNeasy Blood & Tissue kit (QIAGEN, Beijing, China) following the manufacturer’s instructions. DNA was stored at −20 °C until molecular analyses. Unlike the previous studies (e.g., [18]), the gene fragments were extracted from the complete mitogenomes, not sequenced separately. The mitogenomes were sequenced at Berry Genomics, Beijing, China, using an Illumina Novaseq 6000 platform (Illumina, San Diego, CA, USA) with 150 bp paired-end reads. High-quality reads were assembled using an iterative De Bruijn graph de novo assembler, the IDBA–UD toolkit [19], with a similarity threshold of 98% and k values of 40–160 bp. The gene COI was amplified through polymerase chain reaction using universal primers (Appendix A, [20]) as ‘reference sequence’ to target mitochondrial scaffolds and acquire the best fit using IDBA–UD, which achieved a minimum of 98% similarity. Clean reads were manually mapped to the produced mitochondrial scaffolds to specify the precision of the assembly using Geneious 2019.2 [21]. The assembled mitogenome sequences were manually annotated and adjusted using Geneious 2019.2 [21] with a sequence of *Platerodrilus* sp. as a reference [22]. Protein-coding genes (PCGs) were determined by identifying open reading frames, and the secondary structure and positions of 22 tRNAs were predicted using MITOS WebServer (http://mitos.bioinf.uni-leipzig.de/index.py, accessed on 19 September 2022) [23]. Before *cox1*, *cox2*, *nad5*, and *Lrna* were extracted using Geneious 2019.2 [21], the four gene sequences were checked by comparing with sequences from GenBank using a Blast search. The gene fragments of the two *Mesolycus* species were deposited at the NCBI database with accession nos. OP735348-49, OP729892-93, OP803896-99 (Appendix A).

### 2.4. Phylogenetic Analyses

A total of 32 species of the Lycinae were chosen as ingroups, and one species from Lyropaeinae was selected as an outgroup (Appendix A). Except the newly sequenced gene fragments of two *Mesolycus* species, all others were downloaded from the GenBank (Appendix A, [4,5,18,24,25,26,27]). Four mitochondrial gene markers (*cox1*, *cox2*, *nad5*, and *Lrna*) were aligned using MAFFT v7.313 [28] with the default strategy integrated in PhyloSuite v1.2.2 [29]. After removing gaps and ambiguous sites using trimAI with default parameters [30], the separately aligned fragments were concatenated using PhyloSuite v1.2.2 [29] in a combined dataset. The optimal partition schemes and substitution models for the combined dataset were predicted using PartitionFinder2 [31] with the “greedy” algorithm. The “link” estimated branch lengths and the Bayesian information criterion were applied (Appendix A).

Phylogenies were deduced by Bayesian inference (BI) and maximum likelihood (ML) methods. IQ-Tree v.1.6.8 [32] and MrBayes v3.2.6 [33] were employed for ML and BI analyses, respectively. An ML tree was constructed with ultrafast 5000 bootstrap replicates. The BI tree was produced with four chains of two parallel runs for 10 million Markov Chain Monte Carlo (MCMC) generations and was terminated with an average standard deviation of split frequencies < 0.01, in which trees were sampled every 1000 generations. The initial 25% of sampled trees were discarded as burn-in, and the majority rule consensus tree was computed to estimate Bayesian posterior probabilities of each node. Phylogenetic trees were visualised and annotated using FigTree 1.4.4 [34].

## 3. Results

### 3.1. Descriptions of the Species


***Mesolycus rubromarginatus* Kazantsev, 2013**


*Mesolycus rubromarginatus* Kazantsev, 2013 [13]: 249.

Figure 1A–C,G.

**Material examined.** China—2♂♂ (IZAS), Yunnan, Menglongbanna, Mengsong, 1600 m, 23.IV.1958, S. Y. Wang leg.; 1♀ (IZAS), same locality, 26.IV.1958, Y. R. Zhang leg.

**Descriptive notes.** Male. Phallus (Figure 1A,B) stout and uniformly sclerotized, in a horizontal line with apical hood (Figure 1C). Internal sac relatively large, about half length of phallus (Figure 1A–C). Internal sac absent with upper plate, lower plate relatively large and complex, with four conspicuous teeth on dorsal side, apical ring sub–circular (Figure 1G).

**Distribution.** N. Laos, China (new record: Yunnan).

**Remark.** This species is recorded from China for the first time. The male habitus was provided by Kazantsev [13], so it is not present here. The aedeagus is provided with the macrophotographs for the first time to better know its characters and make comparison with others.


***Mesolycus dentatus* Y. Yang, Liu et X. Yang, sp. n.**


Figure 1D–F,H and Figure 2A

**Type material. Holotype**: ♂ (IZAS), China, Yunnan, Menglongbanna, Mengsong, 1600 m, 24.IV.1958, X. W. Meng leg.

**Description. Male** (Figure 2A). Length 7.1 mm, width at humeri 1.5 mm.

Body brown. Pronotum reddish brown and brown in lateral margin of disc. Elytra brick red, mixed with dark patches. Surface covered with sparsely inconspicuous red pubescence.

Head medium size, as wide as anterior margin of pronotum. Vertex flat, without median line. Eyes relatively large, interocular distance about 1.1 times greater than eye diameter. Antennae serrate, overlapping two-thirds length of elytra when inclined. Antennomeres II compressed, shorter than wide of apices, III-X subequal in length, XI longest, nearly parallel-sided, pointed at apex.

Pronotum trapezoidal, 0.7 times as long as wide, disc present with a median longitudinal keel extending from anterior margin to middle part. Anterior margin round, lateral margins diverging posteriorly, posterior margin bisinuate; anterior angles confluent with anterior margin and posterior angles sharply projecting. Scutellum trapezoidal, feebly emarginate at apex. 

Elytra slender, 4.4 times longer than humeral width, with outer margins subparallel, inner margins dehiscent feebly behind the middle. Elytral costae I weak but complete and visible, II and III strong.

Phallus (Figure 1D,E) slender and uniformly sclerotized, at an angle of 150 degrees with apical hood (Figure 1F). Internal sac relatively large, about half length of phallus (Figure 1D–F). Upper plate (Figure 1F) longer than lower plate, almost straight basally, with internal margin largely emarginate and external margin simple. Lower plate (Figure 1F) short, present with a stout ventro-apical process, having four teeth but without laminae on apical surface, of which ventral teeth obtuse, while dorsal ones sharp at apices (Figure 1H).

**Female.** Unknown.

**Distribution.** China (Yunnan).

**Diagnosis.** The new species resembles *M. nanensis* Kazantsev, 2004 in the shape of aedeagus, but differs from the latter in the internal sac, of which lower plate is wider than upper plate, and ventral teeth of lower plate are obtuse, while dorsal ones are sharp (Figure 1H). In contrast, in the internal sac of *M. nanensis*, upper plate and lower plate are equal in width, dorsal teeth of lower plate are truncate, while dorsal ones are sharp at apices ([9]: Figures 44 and 45).

**Etymology.** The specific name is derived from the Latin *dentatus* (toothed, having teeth), referring to its lower plate of inner sac of aedeagus with four teeth.


***Mesolycus qinlinganus* (Kazantsev, 2000)**


*Dilophotes qinlinganus* Kazantsev, 2000 [8]: 330; Bic, 2002 [10]: 5.

*Mesolycus qinlinganus*: Kazantsev, 2004 [9]: 19.

Figure 1K–M,I, and Figure 2B.

**Material examined.** China—3♂♂5♀♀ (IZAS), Gansu, Wenxian, Qiujiaba, 2000–2100 m, 1. VII. 1998; S. Y. Wang leg.; 4♂♂4♀♀ (IZAS), same data as the proceeding, W.Y. Zhou leg.; 3♂♂4♀♀ (IZAS), Gansu, Zhouqu, Shatan Forestry, 2400 m, 17. VII. 1999; H. J. Wang leg.; 1♂1♀ (IZAS), Gansu, Wenxian, Fanba, 800 m, 25. VI. 1998; X. K. Yang leg.; 5♂♂5♀♀ (MHBU), Ningxia, Jingyuan, Erlonghe Forestry, X. P. Wang, H. F. Ran & Q. Q. Wu leg.

**Descriptive notes**. **Male** (Figure 2B). Phallus (Figure 1K,L) slender and uniformly sclerotized, at an angle of 150 degrees with apical hood (Figure 1M). Internal sac relatively large, about half length of phallus (Figure 1K–M). Upper plate shorter than lower plate (Figure 1M). Upper plate (Figure 1M) curved basally, with internal margin largely emarginate and external margin simple. Lower plate (Figure 1M) widely curved ventrally, external margin slightly curled dorsally, present with a slender ventro-apical process, having a pair of modified laminae merged together in middle of apical surface (Figure 1I).

**Distribution.** China (Ningxia, Gansu, Sichuan, Shaanxi, Hubei, Yunnan).

**Remark.** This species is widely distributed, and its coloration of pronotum and elytra has obvious geographical differences. As noted by Bic [10], they are either red or dark red individuals. It is the first time this species has been recorded in Gansu province, and all of the specimens are uniformly dark red both at low and high altitudes. The pronotum and elytra are always concolor.


***Mesolycus shaanxiensis* Y. Yang, Liu et X. Yang, sp. n.**


Figure 1N–P,J and Figure 2C.

**Type material. Holotype:** ♂ (IZAS), China, Shaanxi, Ningshan, Huoditang, 18.VIII.1998, 1580 m, D. C. Yuan leg. Paratype: 1♂ (IZAS), Shaanxi, Zhouzhi, Houzhenzi, 23.VI.1999, 1320 m, Y. W. Zhang leg.

**Description. Male** (Figure 2C). Length 6.3–6.5 mm, width at humeri 1.3 mm.

Body brown to black. Pronotum reddish brown and black in center of disc. Elytra brick red, and all margins bright red. Surface covered with sparsely inconspicuous red pubescence, which is brown on middle of pronotal disc and mixed with long black hairs on antennae.

Head small, narrower than anterior margin of pronotum. Vertex flat, without median line. Eyes relatively small, interocular distance 1.3 times greater than eye diameter. Antennae serrate, overlapping two-fifths length of elytra when inclined. Antennomeres II compressed, shorter than wide of apices, III–X subequal in length, XI longest and tapered apically.

Pronotum trapezoidal, 0.7 times as long as wide, disc present with a median longitudinal keel extending from anterior margin to middle part. Anterior margin extremely round, lateral margins obliquely diverging posteriorly, posterior margin straight; anterior angles confluent with anterior margin and posterior angles moderately projecting. Scutellum trapezoidal, feebly emarginate at apex. Elytra slender, 3.9 times longer than humeral width, with outer margins slightly diverging posteriorly, inner margins dehiscent behind the middle. Elytral costae I weak but complete and visible, II and III strong.

Phallus (Figure 1N,O) slender and uniformly sclerotized, at an angle of 150 degrees with apical hood (Figure 1P). Internal sac relatively small, about one-third the length of phallus (Figure 1N–P). Upper plate (Figure 1P) as long as lower plate, almost straight basally, with internal margin feebly emarginate and external margin simple. Lower plate (Figure 1P) slightly curved ventrally, present with a stout ventro-apical process, having a pair of simple laminae separated on both sides of apical surface (Figure 1J).

**Female.** Unknown.

**Distribution.** China (Shaanxi).

**Diagnosis.** This new species is most alike *M. hubeicus* Kazantsev, 2004, but differs from the latter in the lower plate of internal sac, whose ventro-apical process is stout and present with a pair of simple laminae separated on both sides of apical surface (Figure 1J). In contrast, the ventro-apical process of *M. hubeicus* is slender and absent with laminae on apical surface ([9]: Figure 43).

**Etymology.** The specific name is derived from the type locality, Shaanxi prov., China.


***Mesolycus varus* Y. Yang, Liu et X. Yang, sp. n.**


Figure 2D and Figure 3A–C,J.

**Type material. Holotype:** ♂ (IZAS), China, Yunnan, Weixi, Pantiange, 2800 m, 21.VII.1981, S. Y. Wang leg. Paratypes: 1♂1♀ (IZAS), same as the holotype.

**Description. Male** (Figure 2D). Length 6.5–6.8 mm, width at humeri 1.5 mm.

Body brown to black. Pronotum red, with a square black patch in the center of the disc. Elytra reddish brown and all margins bright red. Surface covered with decumbent red pubescence.

Head relatively small, narrower than anterior margin of pronotum. Vertex flat, with median line. Eyes relatively small, interocular distance 1.5 times greater than eye diameter. Antennae serrated, overlapping elytral mid–length when inclined. Antennomeres II compressed, shorter than wide of apices, III teardrop–shaped, IV–X subequal in length, XI longest, parallel–sided, pointed at apex.

Pronotum trapezoidal, 0.7 times as long as wide, disc present with a median longitudinal keel extending from anterior margin to middle part. Anterior margin extremely round, lateral margins diverging posteriorly and posterior margin feebly bisinuate; anterior angles confluent with anterior margin and posterior angles sharply projecting. Scutellum trapezoidal, feebly emarginate at apex.

Elytra slender, 3.8 times longer than humeral width, outer margins subparallel and inner margins dehiscent behind the middle. Elytral costae I weak and reaching to two-thirds length of elytra, II and III strong.

Phallus (Figure 3A,B) relatively stout and uniformly sclerotized, at an angle of 120 degrees with apical hood (Figure 3C). Internal sac relatively large, about half length of phallus (Figure 3A–C). Upper plate (Figure 3C) as long as lower plate, curved basally, feebly emarginate at internal margin and external margin simple. Lower plate (Figure 3C) slightly curved ventrally, absent with ventro-apical process or teeth, external margin obviously curled dorsally (Figure 3J).

**Female.** Similar to male, but antennae close to filamentous.

**Diagnosis.** This species is different from all others of *Mesolycus* by the aedeagus, whose lower plate is simple and absent with ventro-apical process (Figure 3J), in contrast with other species of which the lower plate is present with ventro-apical process.

**Etymology.** The specific name is derived from the Latin *varus* (bent outwards, turned awry), referring to its internal sac with upper plate obviously curled dorsally.

**Distribution.** China (Yunnan).


***Mesolycus breviplatus* Y. Yang, Liu et X. Yang, sp. n.**


Figure 2E and Figure 3D–F,K.

**Type material. Holotype:** ♂ (IZAS), CHINA, Yunnan, Dimaluo-Biluoxueshan, 6.VII.2019, Y. D. Chen & Q. L. Lei leg. Paratype: 1♀ (IZAS), same data to the holotype.

**Description. Male** (Figure 2E). Length 5.2 mm, width at humeri 0.8 mm.

Body brown to black. Pronotum and elytra orange. Surface covered with dense and decumbent orange pubescence.

Head small, narrower than anterior margin of pronotum. Vertex flat, with median line. Eyes relatively small, interocular distance about 1.7 times greater than eye diameter. Antennae serrated, overlapping two-thirds length of elytra when inclined. Antennomeres II compressed, shorter than width of apices, III–X sub-equal in length, XI longest and parallel-sided, pointed at apex.

Pronotum trapezoidal, about half as long as wide, disc present with a median longitudinal keel extending from anterior margin to three-fourth part. Anterior margin round, lateral margins diverging posteriorly and posterior margin straight; anterior angles round and posterior angles moderately projecting. Scutellum trapezoidal, obviously emarginate at apex.

Elytra slender, 4.1 times longer than humeral width, with outer margins slightly diverging posteriorly, inner margins dehiscent along whole length. Elytral costae I very weak, extending to mid-length of elytral, II stronger than III.

Phallus (Figure 3D–E) slender and uniformly sclerotized, at an angle of 135 degrees with apical hood (Figure 3F). Internal sac relatively large, about half length of phallus (Figure 3D–F). Upper plate (Figure 3F) as long as lower plate, curved basally, with internal margin largely emarginate and external margin simple. Lower plate (Figure 3F) slightly curved ventrally, present with a relatively stout ventro-apical process, having a pair of modified laminae merged together in middle of apical surface (Figure 3K).

**Female.** Similar to male, but body longer and wider.

**Distribution.** China (Yunnan).

**Diagnosis.** This species is most similar to *M. vitalisi* in the structures of aedeagus, but differs from the latter by the lower plate whose external margin is slightly curled dorsally and the upper plate whose external margin is simple (Figure 3K). In contrast, the lower plate of *M. vitalisi* whose external margin is obviously curled dorsally and external margin of the upper plate is concave in the middle ([9]: Figures 38 and 39).

**Etymology.** The specific name is derived from the Latin *brevis* (short) and *plata* (plate), referring to its relatively short lower plate of internal sac of aedeagus.


***Mesolycus murzini* Kazantsev, 2004.**


*Mesolycus murzini* Kazantsev, 2004: 18 [9].

Figure 2F and Figure 3G–I,L.

**Material examined.** CHINA– 5♂♂5♀♀ (MHBU), Xizang, Zayu, Shang Zayu, 14.IV. 2015, A. M. Shi leg.

**Descriptive notes. Male** (Figure 2F). Phallus (Figs 3G,H) slender and uniformly sclerotized, membranous at subterminal margin, at an angle of 135 degrees with apical hood (Figure 3I). Internal sac relatively large, about mid-length of phallus (Figure 3G–I). Upper plate (Figure 3I) longer than lower plate, almost straight basally, feebly emarginate at internal margin, present with two dorso-apical teeth and an elongated bifurcated protrusion in the middle. Lower plate (Figure 3I) curved, present with two ventro-apical teeth, but without ventro-apical process and laminae (Figure 3L).

**Distribution.** Myanmar, China (new record: Xizang).

**Remarks.** This species is recorded from China for the first time. Although the aedeagus of this species was illustrated in the original publication [9], the male habitus and aedeagus are provided with the macrophotographs here for the first time to better know its characters and make comparisons with others.

### 3.2. Taxonomic Notes


***Mesolycus atrorufus* group**


**Remarks.** Bic [8] proposed the *atricollis* group to denote an aggregate of species characterized by serrate male antennae and a slender phallus with a characteristically sclerotized internal sac. However, Kazantsev [9] applied the *atrorufus* group for this aggregate’s name. The Principle of Priority applies to such names, according to the Article 23.3.3 of ICZN [35]. Due to *M. atrorufus* (Kiesenwetter, 1879) being the oldest one in this aggregate of species, we adopted the *atrorufus* group as the valid name in the present study.


***Mesolycus ilyai* group**


**Remarks.** Bic [10] proposed the *moxiensis* group to denote the other aggregate of *Mesolycus* species characterized by the shape of the internal sac, which has the form of a complex plate armed with processes and bent margins. However, *Dilophotes moxiensis* Bic, 2002 was synonymized with *M. ilyai* (Kazantsev, 2000) by Kazantsev [9]. Accordingly, the name of this aggregate of species should be changed into the *ilyai* group, in terms of the Principle of Priority, Article 23.3.5 of ICZN [35].

***Mesolycus atrorufus*** (Kiesenwetter, 1879).

*Eros atrorufus* Kiesenwetter, 1879: 305 [36].

*Mesolycus puniceus* Gorham, 1883: 399 [37]; Kleine, 1933: 6 [14]; Kazantsev, 2004:10. [9].

*Dilophotes atrorufus*: Kleine, 1942 [38]; Nakane, 1969 [6]; Bocak & Bocakova, 1988: 444 [12]; Bic, 2002: 8 [10]; Kazantsev, 2000: 329 [8].

*Mesolycus atrorufus*: Lewis, 1895: 407 [39]; Kazantsev, 2004: 12 [9].

*Dilophotes atrorufus*: Bocakova & Bocak, 2007: 212 [40].

*Mesolycus atrorufus*: Matsuda, 2012:1 [41].

**Distribution.** Japan.

**Remarks.** *Mesolycus puniceus* Gorham, 1883 was synonymized with *M. atrorufus* (Kiesenwetter, 1879) by Kazantsev [9]. However, this taxonomic change was neglected by Bocakova & Bocak [40] in the Palaearctic Catalogue. Besides, the citation of this species was confusing in the past decades, either included in *Dilophotes* or *Mesolycus* by different taxonomists. It definitely belongs to *Mesolycus* (sensu Kazantsev [9]) due to its symmetrical aedeagus including median lobe and internal sac [10].


***Mesolycus atricollis* (Pic,1926) comb. n.**


*Dilophotes atricollis* Pic, 1926: 68 [42]; Bic, 2002: 8 [10]; Bocakova & Bocak, 2007: 212 [40].

**Distribution.** China (Taiwan).

**Remarks.** Based on the similarity of its aedeagus to *M. qinlinganus* [10], we know that this species is a member of *Mesolycus* (sensu Kazantsev [9]).


***Mesolycus particularis* (Pic, 1928) comb. n.**


*Dilophotes particularis* Pic, 1928: 32 [43]; Bic, 2002: 13 [10].

**Distribution.** Vietnam, Nepal.

**Remarks.** Based on the similarity of its aedeagus to *M. qinlinganus* [10], we suggest it be attributed to *Mesolycus* (sensu Kazantsev [9]).


***Mesolycus pacholatkoi* (Bic, 2002) comb. n.**


*Dilophotes pacholatkoi* Bic, 2002: 17 [10].

**Distribution.** China (Yunnan, Guizhou), N. Vietnam, N. Laos, NE. Thailand

**Remarks.** The symmetrical aedeagus [10] makes it as a member of *Mesolycus* (sensu Kazantsev [9]). However, this species may have been neglected by Kazantsev [9] when he redefined *Mesolycus*, although the remaining species were clearly indicated with generic changes (comb. nov.).

### 3.3. Distribution Pattern

A total of 30 *Mesolycus* species are known at present, of which 70% (21 species) belong to the *atrorufus* group, and the remainder (9 species) to the *liyai* group. Generally, the species of the *atrorufus* group and the *ilyai* group overlap in their distribution, but the range of the former is much larger than that of the latter.

In overall view, the *Mesolycus* species are widely distributed in the East Palaearctic and Oriental regions, −0.40–45.20° N, 85.32–147.60° E (Figure 4). Within this region, 13 species are distributed in China (10 species are endemic), accounting for 43.33% of the total number worldwide; six species in Laos; four species in Borneo; three species in Vietnam; and one or two species in the other countries (including Japan, Bhutan, India, Nepal, Myanmar, Thailand, Malacca and Sumatra), respectively.

The species are narrowly distributed, and most of them are restricted to one or a few localities, except for *M. pacholatkoi* (occurring in SW China, N. Laos, N. Thailand and N. Vietnam) and *M. qinlinganus* (restricted to China) covering a relatively large distribution range. Even some species are sympatric, such as *M. hubeicus* and *M. qinlinganus* (China, Hubei prov., Dashennongjia Natur. Res.), *M. dentatus* sp. n. and *M. rubromarginatus* (China, Yunnan prov., Mengsong), *M. bolavensis* and *M. jendeki* (Laos, Attapu prov., Bolvan Plataeu, Nong Lom lake env.), *M. holzschuhi* and *M. bhutanensis* (Bhutan, Thimphu Distr., Menshunang).

### 3.4. Phylogenetic Analyses

Both ML and BI analyses produced almost congruent topologies (Figure 5). The monophyly of all tribes (each with two representative species) as morphologically defined [1,16] was recovered in both analyses, and each of them was well supported (BS = 84–100, PP = 0.98–1). However, the relationships among the tribes of Lycinae got low support (BS = 22–72, PP = 0.31–0.81).

Macrolycini (*Macrolycus*) was a recovered sister to the remaining tribes except for Metriorrhynchini (*Broxylus* and *Cautires*) and Dihammatini (*Dihammatus*), albeit with low supporting values (BS = 45, PP = 0.31). *Mesolycus* was solidly grouped with *Dilophotes* into a monophyletic clade (BS = 100, PP = 1), which is recovered as a sister to Slipinskiini (Flagrax), and had moderate supports (BS = 72, PP = 0.81).

## 4. Discussion 

### 4.1. Separate Status of the New Species

The variability in general appearance sometimes prevents reliable identification, and therefore the delineations of species are regularly based on the shape of the male genitalia, which are provided here in the key. As noted, *M. pygmaeus* Waterhouse, 1879 (Borneo) and *M. ater* Pic, 1943 (Malacca) were excluded from the key because only female types [9] are hitherto known and because of a poor description in the original publications.
**Key to the Males of *Mesolycus* Species**Phallus very slender, internal sac attached to the phallus by two basal rods and additionally formed by two apical sclerotized complex plates (the *atrorufus* group, e.g., Figure 1D–F,H–P and Figure 3A–L)2-Phallus robust, considerably shorter, internal sac attached to the phallus only by a complex stiffness plate (the *ilyai* group, e.g., Figure 1A–C,G)202.Phallus at least three times longer than internal sac (e.g., Figure 1N–P)3-Phallus at most twice longer than internal sac (e.g., Figure 3A–C)63.Phallus curved apically ([9]: Figures 40 and 41), Japan*M. mediozonatus* (Nakane, 1955)-Phallus almost straight44.Lower plate of internal sac straight basally ([10]: Figure 13), Bhutan, SE India*M. holzschuhi* (Bic, 2002)-Lower plate of internal sac curved extensively55.Ventro-apical process of lower plate stout (Figure 1J), China (Shannxi)*M. shaanxiensis* sp. n.-Ventro-apical process of lower plate slender ([9]: Figure 43), China (Hubei)*M. hubeicus* Kazantsev, 20046.The upper and lower plates subequal in length (e.g., Figure 3A–F)7-The upper and lower plates obviously different in length (e.g., Figure 1D–F,K–M and Figure 3G–I)117.Lower plate of internal sac simple, without ventro-apical process (Figure 3J), China (Yunnan)*M. varus* sp. n.-Lower plate of internal sac present with ventro-apical process (e.g., Figure 1H,I and Figure 3K)88.Upper plate of internal sac dentate at external margin (e.g., [10]: Figures 12 and 15)9-Upper plate of internal sac simple at external margin (e.g., Figure 3K)109.Dorso-apical part of external margins of upper plate round ([10]: Figure 15), N. Laos*M. sausai* (Bic, 2002)-Dorso-apical part of external margins of upper plate straight ([10]: Figure 12), Bhutan*M. bhutanensis* (Bic, 2002)10.Lower plate of internal sac wider than upper plate ([9]: Figures 38 and 39), Vietnam*M. vitalisi* (Pic, 1923)-Lower plate and upper plate of internal sac subequal in width (Figure 3K), China (Yunnan)*M. breviplatus* sp. n.11.Upper plate of internal sac longer than lower plate (e.g., Figure 1D–F and Figure 3G–I)12-Upper plate of internal sac shorter than lower plate (e.g., Figure 1K–M)1512.Internal sac almost as wide as phallus (e.g., Figure 3G–I)13-Internal sac obviously wider or narrower than phallus (e.g., Figure 1D–F)1413.Upper plate of internal sac narrow, with elongate narrow upper portion (Figure 3G–I), Myanmar, China (Xizang)*M. murzini* Kazantsev, 2004-Upper plate of internal sac wide, moderately curved apically ([10]: Figure 11), Japan*M. atrorufus* (Kiesenwetter, 1879)14.Upper plate of internal sac and lower plate equal in width, dorso–apical teeth of lower plate truncate at apices ([9]: Figures 44 and 45), Thailand*M. nanensis* Kazantsev, 2004-Lower plate of internal sac wider than upper plate, dorso–apical teeth of lower plate sharp at apices (Figure 1H), China (Yunnan)*M. dentatus* sp. n.15.Lower plate of internal sac twice length of upper plate (e.g., [10]: Figure 14)16-Lower plate of internal sac slightly longer than upper plate (e.g., Figure 1I)1716.Internal margin of lower plate of internal sac dentate ([8]: Figure 5), China (Gansu)*M. berezowskii* (Kazantsev, 2000)-Internal margin of lower plate of internal sac simple ([10]: Figure 14), Laos*M. laosensis* (Bic, 2002)17.Pronotum dark red with infuscate middle, elytra dark red, and sometimes slightly infuscate basally, Vietnam, Nepal*M. particularis* (Pic, 1928) comb. n.-Pronotum concolour, black or red1818.Pronotum black, with very short inconspicuous brown pubescence, elytra brightly orange red, China (Taiwan)*M. atricollis* (Pic, 1926) comb. n.-Pronotum and elytra concolour, dark red to brownish black, and always covered with red pubescence1919.Lower plate of internal sac with a pair of modified laminae merged together on apical surface of ventro-apical process [Figure 1I], China (Sichuan, Shaanxi, Hubei, Yunnan, Gansu, Ningxia)*M. qinlinganus* (Kazantsev, 2000)-Lower plate of internal sac with a pair of simple laminae separate on apical surface of ventro-apical process ([8]: Figure 3), China (Gansu, Yunnan)*M. tibetanus* (Kazantsev, 2000)20.Tarsomeres I and II absent with plantar pads*M. shelfordi* (Bourgeois, 1906)-Tarsomeres I and II present with plantar pads2121.Internal sac extends beyond apex of phallus by at least mid-length of phallus (e.g., Figure 1A)22-Internal sac not or slightly exceeding apex of phallus (e.g., [9]: Figure 37)2622.Internal sac slender, as wide as or narrower than the apical part of the phallus ([9]: Figures 32–35), Borneo, Sumatra*M. obscurus* (Pic, 1912)-Internal sac robust, distinctly wider than apical part of phallus2323.Apical ring twice as wide as long ([10]: Figure 18), China (Sichuan)*M. ilyai* (Kazantsev, 2000)-Apical ring nearly as long as wide (e.g., Figure 1B, [10]: Figure 20)2424.Eyes small, interocular distance 1.36 times maximum eye diameter, China (Yunnan, Guizhou), N. Vietnam, N. Laos, NE. Thailand*M. pacholatkoi* (Bic, 2002) comb. n.-Eyes large, interocular distance at most 1.1 times greater than eye diameter2525.Internal sac present with three teeth on ventral side ([13]: Figures 17 and 18), Laos, China (Yunnan)*M. fedorenkoi* Kazantsev, 2013-Internal sac present with four teeth on ventral side (Figure 1G), Vietnam*M. rubromarginatus* Kazantsev, 201326.Internal sac simple on ventral side ([9]: Figures 36 and 37), Borneo*M. discoidalis* (Pic, 1912)-Internal sac with complex structures on ventral side2727.Phallus 3 times longer than internal sac ([10]: Figure 17), Laos*M. jendeki* (Bic, 2002)-Phallus 5 times longer than internal sac ([10]: Figure 16), Laos*M. bolavensis*(Bic, 2002)

### 4.2. Species Diversity of Chinese Mesolycus

The number of the Chinese *Mesolycus* species is increased from six to 13 through this study, which is unexpected. On the world scale, the total species diversity of *Mesolycus* is highest in China, accounting for nearly a half of the total number. In southern China, there are vast mountainous areas and an usually rich spectrum of vegetation types, numerous geographical and ecological isolations [44], thereby accelerating the speciation of *Mesolycus* there.

The endemism of *Mesolycus* in China is also high (ca. 77%). The high endemism may result from a lack of comprehensive taxonomic studies in the group previously. The widely distributed genus throughout China and surrounding countries is evidence enough that these beetles are, in fact, very well capable of flying and dispersing. This makes it particularly necessary for us to conduct extensive and thorough field surveys in order to fully understand the species diversity of *Mesolycus*. Sympatric distribution is another phenomenon in *Mesolycus*, since two or more species occur at the same locality. The highly disparate morphologies in male genitalia may account for effective reproductive isolation among these sympatric congeneric species.

Despite high endemism, the species-rich Chinese fauna has certain faunal connections with adjacent regions, including the Palaearctic part of East Asia, Indochina, Indo-Malaysia, and the southern Qinghai-Tibet Plateau, by sharing several common species or similar species, such as *M. pacholatkoi, M. murzini* and *M. rubromarginatus*. As a transitional zone between Palaearctic and Oriental regions, China plays a critical role in studying the biogeographic fauna. More specifically, Chinese fauna cannot be neglected in tracking the spatial origin and dispersal of *Mesolycus* or higher grades. Thus, the taxonomy of net-winged beetles remains essential, or even a priority, for the future of systematics, biogeography, ecology, and conservation, etc.

### 4.3. Phylogenetic Position of Mesolycus

In the present study, the sampling of *Mesolycus* was included in the molecular phylogenetic analyses for the first time. In terms of our results, *Mesolycus* was suggested to be a sister group of *Dilophotes* and that they together form a monophyletic Dilophotini. This was consistent with the opinion of Bocak & Bocakova [1], while it conflicts with that of Kazantsev [8,9,17], who hypothesized that *Mesolycus* was more closely related to *Macrolycus* attributed to Macrolycini. Although a tentative apomorphy (presence of wedge cell) was presumed for *Dilophotes* [9], some more common characters were found between the latter and *Mesolychus*, which are the diagnosis to define the Dilophotini [1].

Further, the produced topology demonstrated that Dilophotini was rooted in a distant position from Macrolycini, which was congruent with the molecular phylogenetics of Bocak & Bocakova [1] and Masek et al. [18]. In contrast, a sister relationship was recovered between the two taxa by Kazantsev [8,9,17], based on the analyses of morphological data. Kazantsev [9] proposed the reason for this hypothesis was that they shared some morphological features, as opposed to the bulk of the Lycidae, i.e., presence of the coronal and sub-antennal sutures, presence of the median longitudinal pronotal carina, bifid claws of all tarsi and an asymmetrical phallobase. However, only the bifid claws were acknowledged to be unique for them and hypothesized to be their synapomorphy [9]. However, this character was different between the two tribes, with all claws bifid in both sexes of Macrolycini (*Macrolycus* s. l., [11]) or only in males, while tooted basally in females (*Dilophotes* and *Mesolycus*, [9]) of Dilophotini. Furthermore, the male genitalia is usually considered as a more stable and dependable structure in the taxonomy and classification, as well as in inferring the relationships of Lycidae [16]. Dilophotini (including *Dilophotes* and *Mesolycus*) often have considerably sclerotized and exposed internal sacs, which sometimes form a very complex structure, and a long phallobase [10], while Macrolycini (only *Macrolycus* s. l.) has a simple, slender, membranous internal sac with at most its apical part exposed and a small and rounded phallobase [16]. Moreover, they differ in the number of longitudinal elytral costae, present with three (Dilophotini) or four costae (Macrolycini) [1]. Therefore, these differences are probably adequate to separate Dilophotini and Macrolycini in distant positions. Nevertheless, more DNA markers should be explored to confirm this result due to the low supporting values in inferring the relationships among tribes.

## 5. Conclusions

In the present study, the lycid genus *Mesolycus* is reviewed, with an emphasis on the Chinese fauna. As a result, three species are transferred from *Dilophotes*, including *M. atricollis* (Pic, 1926) comb. n., *M. particularis* (Pic, 1928) comb. n. and *M. pacholatkoi* (Bic, 2002) comb. n. Four new species are discovered in China, including *M. shaanxiensis* sp. n., *M. dentatus* sp. n., *M. breviplatus* sp. n. and *M. varus* sp. n. Two species, *M. murzini* Kazantsev, 2004 and *M. rubromarginatus* Kazantsev, 2013, are recorded in China for the first time. It is quite surprising that the number of Chinese *Mesolycus* species is increased from six to 13, which is much higher than other countries in the species diversity. In addition, the aedeagi are photographed for the previously inadequately illustrated species and a key for the identification of the *Mesolycus* species globally is provided. Moreover, the reconstructed phylogenies of Lycinae based on multiple genes (*cox1*, *nad5*, *cox2* and *Lrna*) by both ML and BI analyses show that *Mesolycus* is a consistently recovered sister to *Dilophotes*, and they together form a monophyletic clade which is well supported, suggesting that *Mesolycus* definitely belongs to Dilophotini rather than to Macrolycini of Lycinae. This study is another contribution to the Chinese lycid beetles and helps us better understand the species diversity and phylogenetic position of *Mesolycus*.

## Figures and Tables

**Figure 1 insects-13-01171-f001:**
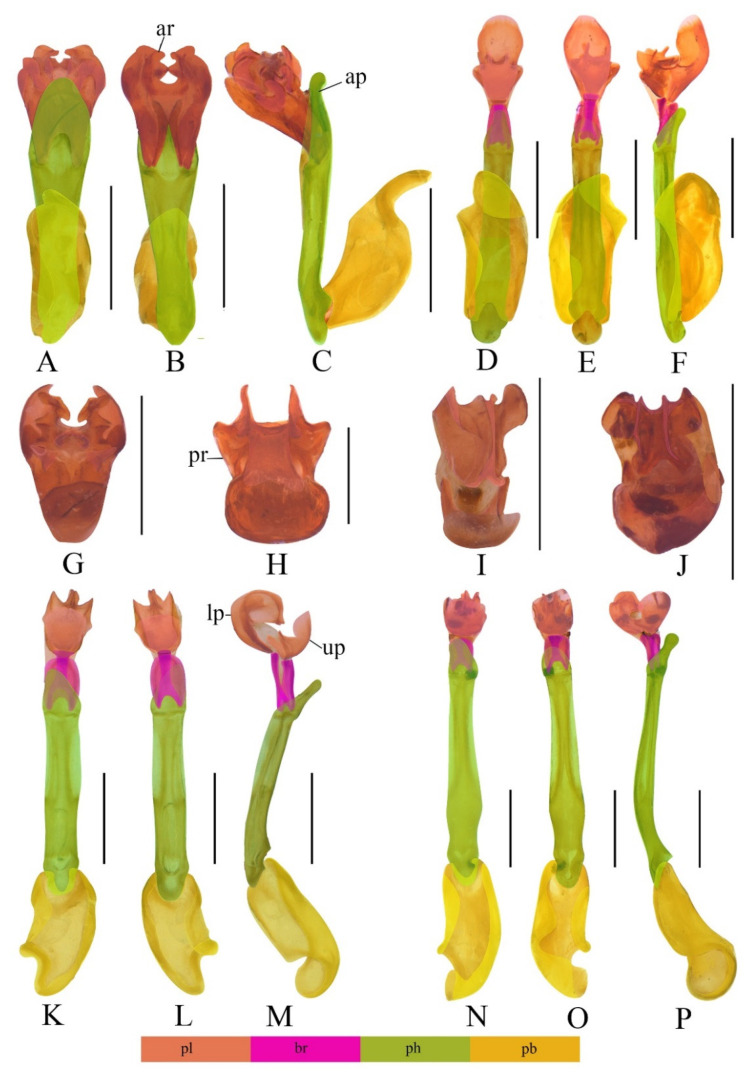
Aedeagi of *Mesolycus* species: (**A**–**C**,**G**). *M*. *rubromarginatus* Kazantsev, 2013; (**D**–**F**,**H**). *M. dentatus* sp. n.; (**K**–**M**,**I**). *M*. *qinlinganus* Kazantsev, 2000; (**N**–**P**,**J**). *M*. *shaanxiensis* sp. n.: (**A**,**D**,**K**,**N**)—ventrally; (**B**,**E**,**L**,**O**)—dorsally; (**C**,**F**,**M**,**P**)—laterally; (**G**–**J**)—apically. Scales: (**A**–**G**,**I**–**P**), 0.5 mm; (**H**), 0.2 mm. pl—plate; br—basal rods; ph—phallus; pb—phallobase; ar—apical ring; ap—apical hood; lp—lower plate; up—upper plate; pr—ventro-apical process.

**Figure 2 insects-13-01171-f002:**
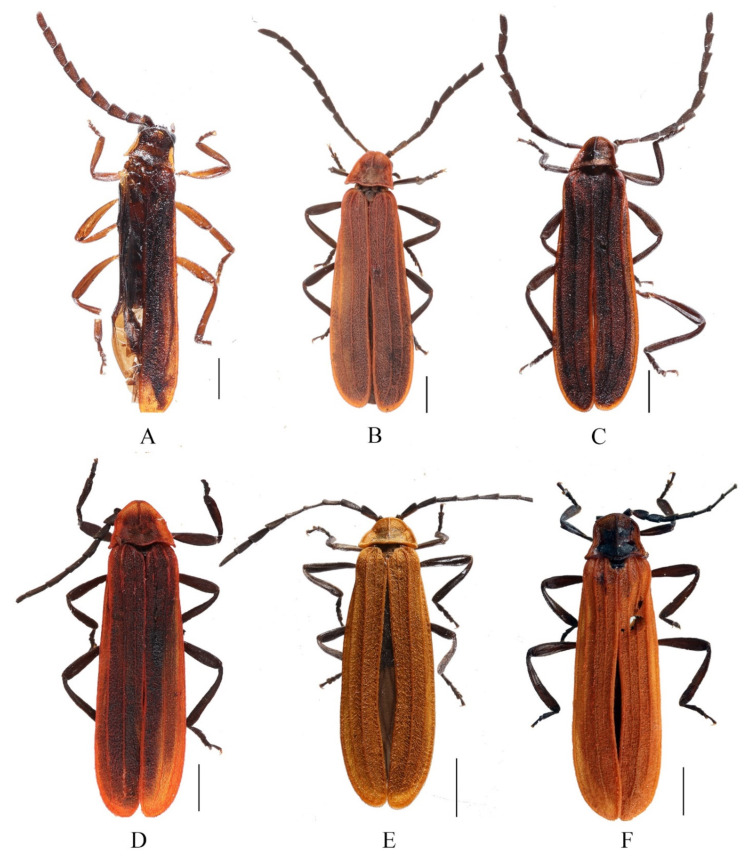
Male habitus of *Mesolycus* species: (**A**). *M. dentatus* sp. n.; (**B**). *M. qinlinganus* (Kazantsev, 2000); (**C**). *M. shaanxiensis* sp. n.; (**D**). *M. varus* sp. n.; (**E**). *M. breviplatus* sp. n.; (**F**). *M. murzini* Kazantsev, 2004. Scales: 1.0 mm.

**Figure 3 insects-13-01171-f003:**
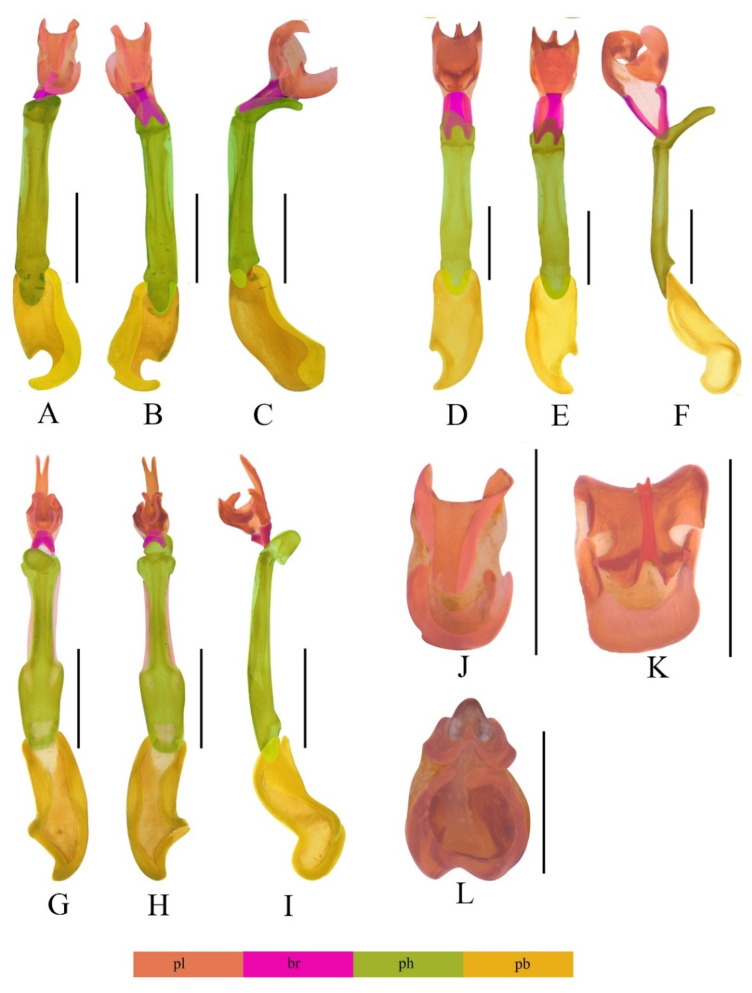
Aedeagi of *Mesolycus* species: *(***A**–**C**,**J**). *M*. *varus* sp. n.; (**D**–**F**,**K**). *M. breviplatus* sp. n.; (**G**–**I**,**L**). *M*. *murzini* Kazantsev, 2004. (**A**,**D**,**G**)—ventrally; (**B**,**E**,**H**)—dorsally; (**C**,**F**,**I**)—laterally; (**J**–**L**)—apically. Scales: (**A**–**I**,**K**), 0.5 mm; (**J**,**L**), 0.2 mm. pl—plate; br—basal rods; ph—phallus; pb—phallobase.

**Figure 4 insects-13-01171-f004:**
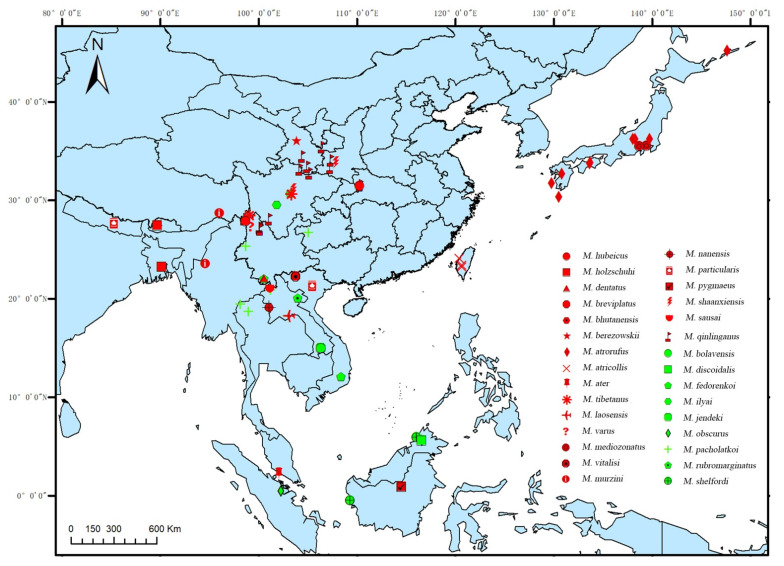
Distribution map of *Mesolycus* in the world scale (The red represents the *atrorufus* group and the green denotes the *ilyai* group).

**Figure 5 insects-13-01171-f005:**
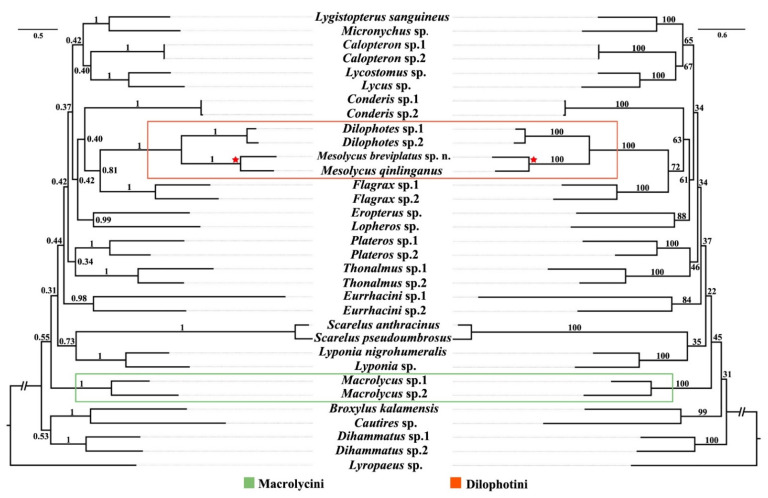
Phylogenetic tree of Lycinae produced from the ML (**right**) and BI (**left**) analyses based on the *cox1*, *nad5*, *cox2* and *Lrna* genes. The numbers on the branches represent bootstrap (BS) (**right**) and posterior probability (PP) (**left**), respectively. Dilophotini is circled in orange and Macrolycini in green box. The pentagram represents *Mesolycus* species used in this study.

## Data Availability

The sequence generated in this study is deposited in GenBank with accession numbers (OP735348-49, OP729892-93, OP803896-99).

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
