# Peer review of "Unexpected High Species Diversity of Mesolycus Gorham (Coleoptera, Lycidae) from China, with a Preliminary Investigation on Its Phylogenetic Position Based on Multiple Genes†"

_insects, 2022, doi:10.3390/insects13121171_

Round 1

Reviewer 1 Report

This is an interesting and needed study on the taxonomy of Mesolycus, a diverse and common group of lycids found in the Palearctic region. The authors present a review of the genus followed by a molecular-based placement of the group within the subfamily Lycinae. While this is an important and overall well done paper, there are some problems that need to be addressed before this manuscript is ready for publication. 

The first one is the language; the text should be more polished and proofread more carefully. There's a lot of little consistent problems found throughout the manuscript that could have been easily picked up if a more thorough language review was done. Some of the sentences are also not very clear and there's a lot of missing or extra words in the whole text. I did the best I could, but it is beyond the scope of my review to check the language of your text.

Second, relates to the fact that the manuscript does not have a single habitus illustration of Mesolycus. Why is this? If the goal of taxonomy is to increase identification rates to stimulate biodiversity preservation, how can you preserve and identify something that you don't know how it looks like? I strongly suggest the inclusion of habitus and other illustrations of all species the authors worked with in the actual main text, not in the supplemental material. The arrangement of genitalia figures is not optimal. Due to the naturally comparative nature of taxonomy, it would be better to have multiple genitalia side by side, to facilitate the comparison between them. Figure 5 is of difficult interpretation, there's too much information and it makes it hard to grasp what is going on; I'd recommend breaking it in two or three maps to make the visualization easier.

In the results section, there are brief diagnostic redescriptions (= descriptive notes) mostly focusing on male genitalia, which are well illustrated, however, the authors failed to reference with figures some other important diagnostic characters listed in this section, especially of the newly described taxa. I made several comments ans suggestions of improvement in the taxonomy part. 

More specifically, in the taxonomic notes section, regarding the use of species groups. According to the ICZN, article 45: The species group encompasses all nominal taxa at the ranks of species and subspecies (see also Article 10.2). See the whole article 45. The authors are apparently erroneously using the term "species group" in their paper, since their application of the term in the text refers to species complex (i.e., multiple species closely related, normally of cryptic nature). This needs to be clarified and properly addressed in the text. 

Lastly, the key, while very much appreciated, needs to be better referenced to images.

Overall a well designed study, but the collection of small errors and proposed changes lead me to suggest this paper needs major revision before it is accepted for publication. See specific comments in the manuscript.

Author Response

Please see the responses in the attached file. 

Reviewer 2 Report

The topic is relevant because it is important to study the biodiversity of East Asia. The history of the study is well described. Descriptions of new species correspond to the Code of Zoological Nomenclature. The conclusions are justified.

My remarks:

It would be nice if the authors could justify such a high diversity by linking it to past events.

It would be good to show the time of divergence of these species.

It would be good to give a key for females.

Authors need to give an image of one male and one female in the article itself.

Authors should give all species and genera names in italics. I showed examples. Everything needs to be fixed.

Small remarks in the text, including words from the title should not be repeated in keywords; and see in the pdf.

Author Response

please see the responses in the attached file. 

Round 2

Reviewer 1 Report

The authors provided satisfactory corrections to all raised questions and concerns provided in a previous version of this manuscript, my recommendation is that this paper is accepted for publication.